# Hybrid Immunity and the Incidence of SARS-CoV-2 Reinfections during the Omicron Era in Frontline Healthcare Workers

**DOI:** 10.3390/vaccines12060682

**Published:** 2024-06-19

**Authors:** Carmen-Daniela Chivu, Maria-Dorina Crăciun, Daniela Pițigoi, Victoria Aramă, Monica Luminița Luminos, Gheorghiță Jugulete, Viorela Gabriela Nițescu, Andreea Lescaie, Cătălin Gabriel Apostolescu, Adrian Streinu Cercel

**Affiliations:** 1Department of Epidemiology 1, Carol Davila University of Medicine and Pharmacy, 050474 Bucharest, Romania; carmen-daniela.chivu@drd.umfcd.ro (C.-D.C.); daniela.pitigoi@umfcd.ro (D.P.); 2Emergency Clinical Hospital for Children “Grigore Alexandrescu”, 011743 Bucharest, Romania; viorela.nitescu@umfcd.ro (V.G.N.); andreea.lescaie@drd.umfcd.ro (A.L.); 3National Institute for Infectious Diseases “Prof. Dr. Matei Balș”, 021105 Bucharest, Romania; victoria.arama@umfcd.ro (V.A.); luminita.luminos@umfcd.ro (M.L.L.); gheorghita.jugulete@umfcd.ro (G.J.); catalin-gabriel.apostolescu@drd.umfcd.ro (C.G.A.); adrian.streinucercel@umfcd.ro (A.S.C.); 4Department of Infectious Diseases 1, Carol Davila University of Medicine and Pharmacy, 020021 Bucharest, Romania; 5Department of Infectious Diseases 3, Carol Davila University of Medicine and Pharmacy, 020021 Bucharest, Romania; 6Department of Pediatrics, Carol Davila University of Medicine and Pharmacy, 020021 Bucharest, Romania

**Keywords:** SARS-CoV-2, reinfection, healthcare workers, vaccines protection, variant of concern

## Abstract

During the coronavirus disease (COVID-19) pandemic healthcare workers (HCWs) acquired immunity by vaccination or exposure to multiple variants of severe acute respiratory syndrome coronavirus 2 (SARS-CoV-2). Our study is a comparative analysis between subgroups of HCWs constructed based on the number of SARS-CoV-2 infections, vaccination, and the dominant variant of SARS-CoV-2 in the population. We collected and analyzed data using the χ^2^ test and density incidence of reinfections in Microsoft Excel for Mac, Version 16.84, and MedCalc^®^, 22.026. Of the 829 HCWs, 70.1% (581) had only one SARS-CoV-2 infection and 29.9% (248) had two infections. Of the subjects with two infections, 77.4% (192) worked in high-risk departments and 93.2% (231) of the second infections were registered during Omicron dominance. The density incidence of reinfections was higher in HCWs vaccinated with the primary schedule than those vaccinated with the first booster, and the incidence ratio was 2.8 (95% CI: 1.2; 6.7). The probability of reinfection was five times lower (95% CI: 2.9; 9.2) in HCWs vaccinated with the primary schedule if the first infection was acquired during Omicron dominance. The subjects vaccinated with the first booster had a density incidence of reinfection three times lower (95% CI: 1.9; 5.8) if the first infection was during Omicron. The incidence ratio in subgroups constructed based on characteristics such as gender, age group, job category, and department also registered significant differences in density incidence. The history of SARS-CoV-2 infection by variant is important when interpreting and understanding public health data and the results of studies related to vaccine efficacy for hybrid immunity subgroup populations.

## 1. Introduction

The COVID-19 pandemic was declared by the World Health Organization (WHO) to be a public health emergency of international concern (PHEIC) on 30 January 2020, based on International Health Regulations (IHR) [1,2]. The meeting of the IHR Emergency Committee on the COVID-19 pandemic on 4 May 2023 acknowledged that this public health emergency was an established and ongoing health issue [3,4]. During the time interval when COVID-19 was PHEIC, almost 7 million deaths and over 750 million confirmed cases were reported [5].

Due to the potential evolution of SARS-CoV-2, the WHO has recommended a long-term strategic attitude consisting of calibrated measures regarding research, public health interventions, and continuous collaborative surveillance [6,7]. Studies and public health indicators have shown a waning immunity against evolving variants of SARS-CoV-2, thus the WHO recommended vaccination and revaccination for priority groups [8]. The effectiveness of prevention and control strategies has also been proposed for monitoring [8]. 

Genomic surveillance of SARS-CoV-2 showed an important genetic diversity, with an impact on transmission, case severity, and immunity escape of the virus. The variants of SARS-CoV-2 were named variants of concern (VOC) if important genetic mutations with an impact on transmission, severity, or immunity escape were acquired, or variants of interest and variants under monitoring in cases of minor mutations.

The laboratory testing of suspected cases and genome sequencing were guided through methodologies, also reporting the data [9,10,11,12]. The Romanian National Institute of Public Health (NIPH) released methodological criteria for the genomic surveillance of SARS-CoV-2 [11]. Changes in the epidemiological, laboratory, and representativeness of the selection criteria are summarized in Appendix A. Reports regarding genomic surveillance were published weekly by the Romanian NIPH, starting from February 2021 [13]. Summarized data regarding the positive cases of SARS-CoV-2 VOC are presented in Appendix A.

In Romania, SARS-CoV-2 evolved from the ancestral variant (hCoV-19/Wuhan/WIV04/2019) to Alpha (B1.1.7 lineage), Delta (B1.1617.2), and Omicron variants [13,14]. The Alpha variant (B1.1.7 lineage) was characterized by an increase in the severity of the cases and the transmission, with no impact on immunity escape. The Delta variant (B1.1617.2) exhibited an impact on immunity escape and a rise in the transmission and severity of cases. All Omicron variants for which evidence-based data existed were characterized by an increase in transmission, a decrease in the severity of the cases, and an impact on immunity protection after vaccination [15,16]. The mutation registered in the Omicron variants involved the spike protein and receptor binding domain. These mutations have affected the transmission dynamics and the targets of the neutralizing antibodies, resulting in a lower efficiency of the previous vaccination. Updated vaccines have been developed to address the decreasing immunity against new Omicron variants. Although the originally authorized vaccines continue to be efficient in preventing hospitalization, severe illness, and death, protection against mild and moderate clinical outcomes is low. 

During the COVID-19 pandemic, HCWs acquired immunity induced by vaccination, SARS-CoV-2 infection, or both. Recommendations for the use of updated vaccines for primary vaccination and re-vaccination have been made for risk groups [8,17]. In populations with hybrid immunity, the SARS-CoV-2 VOC, which determined the prior infection, may play an important role in individual protection and in the incidence of reinfections.

The current study continues and complements previous research on the risk of COVID-19 in HCWs in a designated hospital for treating COVID-19 patients in Bucharest, Romania. The previously published research evaluated the clinical outcome in relation to nonpharmacological protective measures, vaccination, and individual risk [18,19]. Data showed that risk factors associated with the severity of the first SARS-CoV-2 infection were exposure to COVID-19 confirmed cases, age, obesity and anemia [18]. The risk of COVID-19 across all clinical events was lower in HCWs with the first booster administered compared with the risk in HCWs with the primary schedule or who were nonvaccinated [19].

The current study aims to compare the transmission dynamics of SARS-CoV-2 in HCWs vaccinated with the primary schedule and the first booster by estimating the density incidence of reinfections during the Omicron variant circulation in Romania. The study also considers the circulating variant of SARS-CoV-2 during the first infection of HCWs.

## 2. Materials and Methods

### 2.1. Hospital Setting, Participants, and Definitions

During the COVID-19 pandemic in Romania, the National Institute for Infectious Diseases (INBI) “Professor Dr. Matei Balș” in Bucharest admitted multiple COVID-19 cases covering all pandemic waves. HCWs were actively surveyed, and epidemiological investigations were performed on a case-by-case basis, to establish risk factors related to work or individuals.

This retrospective cohort study included HCWs with previously confirmed SARS-CoV-2 infection. The cases were identified and tested according to the national methodology and testing protocol [11]. Cases were suspected if the clinical criteria from the case definition were met, or if there was an identified high risk of exposure in the community or hospital. Laboratory criteria consisted of positive reverse transcription–polymerase chain reaction (RT–PCR) or rapid antigen test (RAT).

Protection of HCWs during medical activities was provided through personal protective equipment (PPE) and vaccination campaigns when vaccines became available. HCWs were trained in the proper use of PPE as a specific regulation of the hospital. The vaccination campaign for the primary schedule started on 27 December 2020, and from September 2021 booster shots were administered. A vaccination center was established at the hospital level and HCWs were prioritized for vaccination.

The HCWs were included in the study based on their professional activity at INBI “Prof. Dr. Matei Balș”, as confirmed by their employment contract, and the history of previous SARS-CoV-2 infection detected according to national methodology [11]. The exclusion criteria were undetermined professional role, unknown infection status before vaccination, and uninfected HCWs. Reinfections were defined as confirmed cases by methodological laboratory testing at least 90 days apart from a previous infection. HCWs with more than two infections were registered; however, they did not meet all the study protocol criteria and were excluded from the analysis. 

The HCWs were considered vaccinated with the primary schedule after 14 days from the last dose administered had elapsed. HCWs vaccinated with the first booster were represented by persons who received an additional dose of the vaccine at least six months after receiving the primary schedule. High-risk departments were defined as wards where there were identified possibilities for direct and indirect transmission by contact with suspected or confirmed COVID-19 cases or their contaminated environments. The healthcare auxiliary role was defined as employees who participated in activities related to the environment of the patients, but without performing any procedures (e.g., cleaning staff). 

Hybrid immunity was defined as the immune protection of vaccinated individuals who had experienced at least one SARS-CoV-2 infection before or after the initiation of vaccination [20]. The history of SARS-CoV-2 infection by variant was assumed based on the circulant variant reported by the NIPH in Romania, with Omicron being dominant from January 2022 (Appendix A) [13]. The study workflow and timeline of the surveyed events are shown in Figure 1.

### 2.2. Study Period

We analyzed the reinfection in HCWs from January 2022 to the end of May 2023, when SARS-CoV-2 Omicron variants were dominant in Romania. 

### 2.3. Data Sources and Collection

The primary data source was the register of HCWs constructed by the Infection Prevention and Control (IPC) team prospectively with data regarding age, gender, professional role, department of activity, laboratory tests, vaccination, and recommendations. Vaccination status was verified retrospectively from the National Electronic Registry of Vaccinations, along with information regarding the date of vaccination [21]. 

### 2.4. Statistical Analysis

The comparison groups were independently constructed based on the number of SARS-CoV-2 infections, primary vaccination schedule, first booster shot, and history of the first SARS-CoV-2 infections during the Omicron era. A descriptive analysis of the study group and the infections and reinfections during SARS-CoV-2 VOC in HCWs was performed. 

The comparative analysis used groups of HCWs without SARS-CoV-2 reinfection and HCWs with reinfections and characteristics such as age, gender, professional role, department, and vaccination status. We also compared groups of HCWs with the first SARS-CoV-2 infection acquired before or after the dominance of SARS-CoV-2 Omicron variants in Romania. We evaluated and compared the density incidence of reinfections in HCWs vaccinated with the primary schedule and those vaccinated with the first booster.

Continuous and categorical variables were presented in mean, median, interquartile ranges (IQR), numbers, and percentages. Differences between the proportions of categorical variables were assessed using χ^2^ tests in MedCalc^®^ 22.026 Software by Ostend, Belgium (https://www.medcalc.org; accessed on 1 March 2024). Density incidence was calculated based on the number of events in different groups of HCWs reported as the number of person-days of HCWs at risk [22]. The results are expressed in values per 10,000 person-days with 95% confidence intervals (95% CI). The comparison of the density incidence was made using the incidence ratio, with a threshold for significance of *p*-value < 0.05. Visualization of data was performed using Microsoft^®^ Excel for Mac, Version 16.84.

### 2.5. Ethics Approval

The Institutional Bioethics Committee approved the study protocol with the registration numbers C02648/16.03.2022 and C08784/27.07.2023. Due to the retrospective nature of the study, only standard informed consent was required from the participants. Access to data was limited to the IPC team and study investigators, ensuring confidentiality.

## 3. Results

We included 829 participants with documented SARS-CoV-2 first infection, of which 70.1% (581) had no reinfection and 29.9% (248) had SARS-CoV-2 reinfection. The study group was composed of 84.3% (699) females with a median age of 44 years, with an interquartile range (IQR) of 36 to 51. Of the female participants, 41.5% (344) were nurses and 71.4% (592) worked in high-risk departments. Comparing the group of HCWs without SARS-CoV-2 reinfection and the group of HCWs with SARS-CoV-2 reinfection, we found statistically significant differences regarding the frequency of the cases by department. In high-risk departments, we identified 77.4% (192) reinfections (*p* = 0.012). Detailed data regarding the descriptive demographics of HCWs from the study and comparison between groups are presented in Table 1.

Regarding vaccination status, the study group had 737 (88.9%) HCWs vaccinated with the primary schedule, 301 (36.7%) vaccinated with the first booster, and 92 (11.1%) HCWs were unvaccinated. Among HCWs vaccinated with the first booster, only one person had an updated vaccine formula. The second booster was identified in 21 (6.9%) persons, of whom 9 (42.9%) were vaccinated with originally authorized vaccines and 12 (57%) had boosters with bivalent vaccines.

Descriptive data regarding the first SARS-CoV-2 infection and the reinfections in different periods of the COVID-19 pandemic defined by the dominance of SARS-CoV-2 variants in Romania, also correlated with data regarding vaccination, are presented in Table 2.

The data indicated that 45.6% (378) of the first SARS-CoV-2 infections were registered during the dominance of the ancestral variant of SARS-CoV-2 and 37.2% (308) during the Omicron variants. There were 248 (29.9%) reinfections, of which 93.2% (231) were detected during the Omicron dominance period. Of the SARS-CoV-2 reinfections during the Omicron period, 56.3% (130) were registered in HCWs vaccinated with the primary series and 31.2% (72) in HCWs who received the first booster. The reinfected HCWs had a median number of days since the last vaccine shot of 365 days (IQR: 238–519) if they were vaccinated with the primary schedule and 169 days (IQR: 99–361) if they acquired the first booster shot.

The vaccination campaign for HCWs commenced on 27 December 2020. The HCWs’ option regarding vaccination with the primary schedule continued until April 2022. The administration of booster shots started in September 2021, and the HCWs’ option regarding the first booster continued until September 2022.

Cumulative data regarding the first and second SARS-CoV-2 infections, vaccination status, and booster are represented graphically in Figure 2. The temporal representation of both the first and second SARS-CoV-2 infections pointed out a dramatic rise in the number of cases starting at the end of 2021, corresponding with the detection of Omicron variant BA.1 of SARS-CoV-2 in the community; in January 2022, the variant becoming dominant.

### 3.1. Reinfections during Omicron in Vaccinated HCWs

Comparing the density of incidence of SARS-CoV-2 reinfections of HCWs vaccinated with the primary series and those vaccinated with the first booster showed that the male HCWs vaccinated with the primary series had a density of incidence of 2.8 (95% CI: 1.2; 6.7) times higher than that of HCWs vaccinated with the first booster. Other groups also registered differences, but with no statistical significance indicated by a *p*-value > 0.05 or 95% confidence interval including a value of 1. Detailed data are presented in Table 3.

### 3.2. Reinfections in HCWs Vaccinated with the Primary Series by the History of SARS-CoV-2 Infection

Overall, there were fewer COVID-19 cases in the group with a history of infection after January 2022. The incidence density of reinfections in HCWs vaccinated with the primary schedule registered significant differences between groups constructed based on the history of infection before and after January 2022. HCWs with a history of COVID-19 before Omicron variant dominance had an incidence density 5.0 (95% CI 2.9; 9.2) times higher than HCWs with a history of COVID-19 after Omicron became dominant in Romania. Subgroups that had significant differences were females, with an incidence density in the group with a history of first COVID-19 before January 2022 that was 6.1 (95% CI 3.2; 13.1) times higher than in the group with a history of first infection after January 2022.

The incidence density by age groups indicated significant differences between age groups above 30 years, with a lower incidence density in the group with a history of infection after January 2022. The job category presented significant differences only for density incidence in the healthcare auxiliary and other categories. In both high-risk and low-risk departments, there were registered differences with lower values in the group of HCWs with a history of infection after January 2022. The data are presented in Table 4.

### 3.3. Reinfections in HCWs Vaccinated with the First Booster and the History of SARS-CoV-2 Infection

Overall, there were fewer COVID-19 cases in the group with a history of infection after January 2022. The incidence density of reinfections in HCWs was significantly different between subgroups of females, age groups over 30 years, nurses, physicians, healthcare auxiliary personnel, and high-risk departments. All mentioned subgroups had a lower incidence density if the participants had a history of SARS-CoV-2 infection after January 2022. Detailed data are presented in Table 5.

## 4. Discussion

The present study identified aspects related to the dynamics of SARS-CoV-2 transmission in HCWs, considering specific characteristics of the participants such as gender, age, occupational category, department of activity, vaccination status, and history of prior SARS-CoV-2 infection before or during Omicron dominance in Romania. 

The first objective of this study was to compare groups to identify differences between HCWs without reinfection versus HCWs with reinfection during the study period. The current study showed that the characteristics represented by gender, age, and occupational category did not indicate statistically significant differences in the comparison of HCWs groups. Reinfections were more common in HCWs from high-risk departments. Our study detected a higher rate of reinfections in HCWs than other studies as the current study comprised a much longer time period and implied multiple variants of Omicron [23,24,25,26,27,28,29,30,31]. Also, in the studies performed during Omicron circulation the populations received multiple vaccine doses after the primary schedule [25,26,30,31].

We considered the circulation of SARS-CoV-2 in the community by establishing periods of dominance of VOC strains according to weekly reports from the Romanian NIPH. Thus, we had four periods: the initial variant of SARS-CoV-2 dominance (hCoV-19/Wuhan/WIV04/2019), and the dominance of VOC Alpha, Delta, and Omicron [13,14].

The dominance periods of SARS-CoV-2 VOC strains varied in length, with a shorter period of dominance circulation of the Alpha and Delta VOC, accounting for almost four months and six months, respectively, during 2021 [13,14]. In Romania, the initial variant of SARS-CoV-2 (hCoV-19/Wuhan/WIV04/2019) circulated for approximately one year, and Omicron variants became dominant at the beginning of 2022. During the Omicron dominance, there was an increase in the number of COVID-19 cases, primary infections, and reinfections in HCWs, although most of the HCWs were vaccinated with the primary schedule, and some also had the first booster administered. The assumption that the high number of infections could be explained by the increased transmissibility of Omicron variants has been demonstrated in multiple studies worldwide [15,16,32].

The time elapsed from vaccination to infection or reinfection is an important indicator for tailoring public health measures regarding the administration of booster shots [33,34,35]. In our study, the average time elapsed from the primary vaccination to reinfection was almost one year. Although a significant proportion of the participants received boosters, there were several reinfections in this category as well. Notably, the boosters used were not adapted to the circulating variants.

Natural and artificial immunity has been assessed by evaluating the dynamics of protective antibodies over time or by using epidemiological indicators from surveillance data. Both types of study have shown that natural and artificial immunity both wane over time and have led to the adaptation of recommendations regarding vaccination schedules [8,17,19,27,35,36,37,38,39,40,41,42]. Common epidemiological indicators of frequency in other studies showed increases in the number of reinfections and primo-infections in the Omicron era, similar to our study [25,26,43,44]. The previously published study on the first SARS-CoV-2 infection, on the same population of HCWs, showed a higher severity of the cases registered in HCWs from the high-risk departments [18].

The demographic characteristics of HCWs, such as age, gender, profession, and department, are important in identifying differences between subgroups. These characteristics have been studied and reported in surveillance data throughout the COVID-19 pandemic, providing information on potential protective measures or additional monitoring of these categories [25,26].

The incidence density of reinfections during Omicron in the male gender and in HCWs from high-risk departments was higher in HCWs vaccinated with the primary schedule than in HCWs vaccinated with the first booster.

A study from the United States of America (USA) conducted in 2020–2021 on a cohort of 62,310 people in the general population also reported a higher frequency of COVID-19 cases in the male gender [45]. Other studies have reported more frequent reinfections in males [46,47].

The higher frequency of infections and reinfections during Omicron circulation reported in other studies has been explained by changes in the transmissibility of Omicron variants and vaccine effectiveness [16,43,44,48]. Although during the study period the HCWs wore PPE in healthcare activities, in the community they were exposed to asymptomatic or suspected COVID-19 cases, as restrictions were fewer during Omicron compared to other waves. The proper use of PPE was assessed for the first COVID-19 confirmed cases in a previously published study using a WHO standardized questionnaire, and a high risk of exposure was identified in 6.8% of HCWs [18].

In terms of disease history, published studies have supported the importance of hybrid immunity [19,39,49]. The study ORCHESTRA compared subjects re-vaccinated with the first booster and a history of SARS-CoV-2 infection with subjects with no prior infection, and the relative risk ratio was 0.11 (95% CI: 0.05–0.25). A cohort retrospective study of HCWs showed a rate of reinfection of 16.2% (95% CI: 13.5–19.3%) in the hybrid immunity group compared with HCWs with no prior infection, which registered a rate of reinfection of 41.7% (95% CI: 40.3–43.0%). In our cohort, surveillance data demonstrated an advantage in terms of protection in individuals with hybrid immunity compared with HCWs vaccinated with the primary schedule and no prior infection, with a hazard ratio (HR) across all COVID-19 events of 0.6 (95% CI: 0.47–0.77) [19]. The HCWs with hybrid immunity and first booster compared to HCWs vaccinated with the primary schedule and no prior infection had a HR of 0.42 (95% CI: 0.30–0.58) [19].

The incidence density of reinfection in different subgroups of HCWs vaccinated with the primary schedule had higher values if there was a history of the first SARS-CoV-2 infection before Omicron compared to HCWs with a history of disease after Omicron dominant circulation. The low likelihood of developing reinfections in this subgroup of HCWs with the first SARS-CoV-2 infection during Omicron emphasizes the importance of the circulating variants when interpreting the hybrid immunity.

Comparing the incidence density of HCWs vaccinated with the primary schedule and the first booster, it was found that those with a history of primoinfection during Omicron were three times less likely to develop a reinfection compared to those who had a history of reinfection before Omicron, but the incidence of reinfections in HCWs vaccinated with the primary schedule versus HCWs vaccinated with the first booster showed no significant differences. This suggests the need for adapted boosters and the importance of recent infections in interpreting vaccine efficacy in populations with hybrid immunity [15,16,50,51,52,53].

Studies demonstrating complex mechanisms or correlating antibody levels and their dynamics postvaccination or post-SARS-CoV-2 infection are extremely useful for understanding the pathological phenomenon. In a hybrid immunity population, the infection-induced and the vaccine-induced antibodies cannot be differentiated, but the duration of persistence of the antibodies is very useful for establishing vaccination programs and the optimal timing for boosters [15,16,20,53,54,55]. As the current study shows, indicators like density incidence are useful for evaluating the effectiveness of control measures in specific subgroups of the population in specific SARS-CoV-2 VOC circulation periods.

The strengths of this study are represented by the quality of data obtained through active surveillance of HCWs by a specialized team in field epidemiology. In addition, the data (comprised of multiple sources) were both manually and automatically validated.

However, our study had some limitations. In particular, we did not consider transition periods where two SARS-CoV-2 variants of concern were co-circulating, and we only took into account the moment when a VOC of SARS-CoV-2 became dominant, a situation imposed given the reduced circulation as a time period of Alpha and Delta variants. However, the objective was to assess the incidence of pre-Omicron disease and post-Omicron disease in HCWs, but not for each strain. In addition, we did not have data to compare non-adapted boosters with adapted boosters, as our cohort had administered only a few adapted boosters.

## 5. Conclusions

The presence of a history of illness during the Omicron circulation period influenced the incidence density of reinfections in HCWs, including in subgroups based on demographic characteristics. Thus, our study highlights the importance of considering the history of SARS-CoV-2 infection when interpreting and understanding the public health data and the results of studies related to the evaluation of vaccine efficacy in hybrid immunity subgroup populations.

## Figures and Tables

**Figure 1 vaccines-12-00682-f001:**
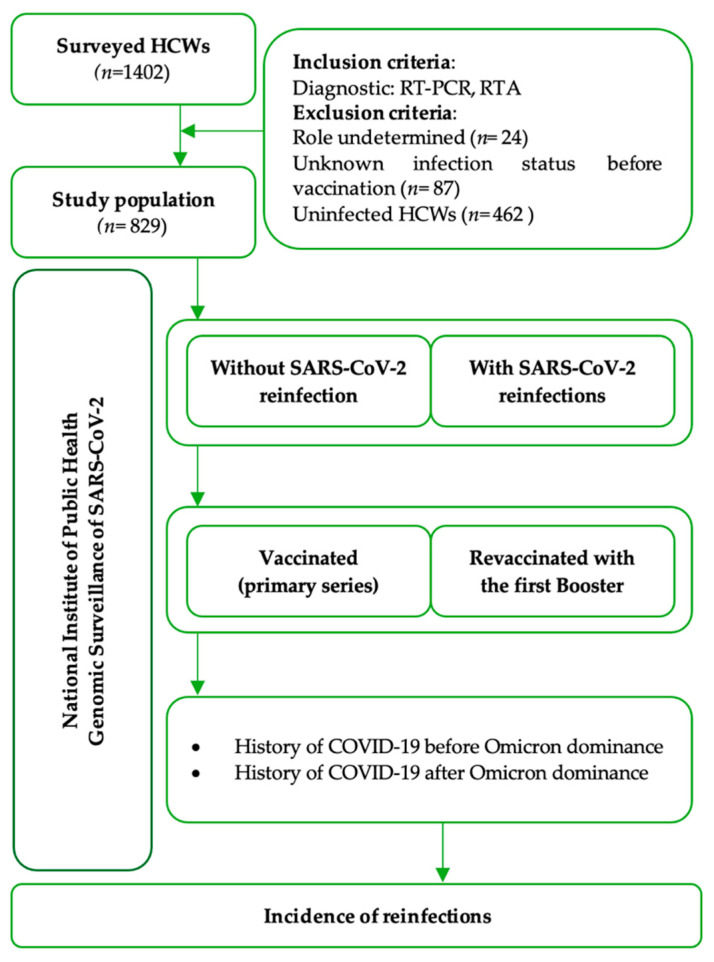
The workflow diagram of the HCWs study in the National Institute for Infectious Diseases “Professor Dr. Matei Balș”, Bucharest, Romania, January 2022–May 2023.

**Figure 2 vaccines-12-00682-f002:**
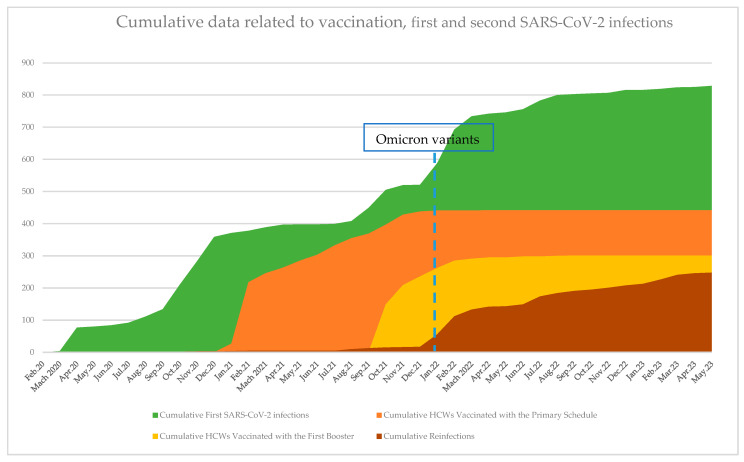
Graphical representation of cumulative data regarding vaccination status, first and second SARS-CoV-2 infections in HCWs included in the study.

**Table 1 vaccines-12-00682-t001:** Descriptive characteristics of the study group and comparative analysis between HCWs without reinfections and HCWs with reinfection during the study period.

Characteristics	Study Participants	Without SARS-CoV-2 Reinfection	With SARS-CoV-2 Reinfection	*p*-Value
	829 (%)	581 (70.1)	248 (29.9)	
**Gender**				
Female	699 (84.3)	491 (84.5)	208 (83.9)	0.828
Male	130 (15.7)	90 (15.5)	40 (16.1)	
**Age**				
Mean (SD)	43 (9.9)	43 (10.2)	43 (9.14)	
Median age (years), IQR	44 (36–51)	44 (35–44)	44 (36–50)	
≤29	78 (9.4)	61 (10.5)	17 (6.9)	0.105
30–39	196 (23.6)	136 (23.4)	60 (24.2)	0.804
40–49	293 (35.3)	200 (34.4)	93 (37.5)	0.393
≥50	262 (31.6)	184 (31.7)	78 (31.5)	0.955
**Job category**				
Nurses	344 (41.5)	236 (40.6)	108 (43.5)	0.438
Physicians	156 (18.8)	111 (19.1)	45 (18.1)	0.736
Healthcare auxiliary activities	203 (24.5)	143 (24.6)	60 (24.2)	0.903
Other categories	126 (15.2)	91 (15.7)	35 (14.1)	0.557
**Department**				
High Risk ^1^	592 (71.4)	400 (68.8)	192 (77.4)	0.012
Low risk ^2^	237 (28.6)	181 (31.2)	56 (22.6)	

^1^ ICU, emergency, COVID-19 wards; ^2^ nonCOVID-19 wards, laboratory; SD, standard deviation; IQR, interquartile range.

**Table 2 vaccines-12-00682-t002:** Descriptive analysis of the first and second SARS-CoV-2 infection of the HCWs from INBI” Prof. Dr. Matei Balș” by pandemic wave corresponding to the predominant transmission period of main SARS-CoV-2 variants.

Characteristics of COVID-19 Cases in the Study Group	First SARS-CoV-2 Infection		Second SARS-CoV-2 Infection	
	829	%	248	29.9 * (%)
**hCoV-19/Wuhan/WIV04/2019 (February 2020–February 2021)**				
**Total**	**378**	**45.6**	**5**	**2.0**
Nonvaccinated HCWs	378	45.6	5	100.0
HCWs vaccinated with the primary series	0	0.0	0	0.0
HCWs vaccinated with the first booster dose	0	0.0	0	0.0
**Alpha (March 2021–June 2021)**				
**Total**	**20**	**2.4**	**0**	**0.0**
Nonvaccinated HCWs	14	70.0	0	0.0
HCWs vaccinated with the primary series	6	30.0	0	0.0
Median number of days since immunization (IQR)	44.5 (36–52)	-	-	-
HCWs vaccinated with the first booster dose	0	0.0	0	0.0
**Delta (June 2021–December 2021)**				
**Total**	**123**	**14.8**	**12**	**4.8**
Nonvaccinated HCWs	38	30.9	6	50.0
HCWs vaccinated with the primary series	85	69.1	6	50.0
Median number of days since immunization (IQR)	240 (218–257)	-	203 (169–301)	-
HCWs vaccinated with the first booster dose	0	0.0	0	0.0
**Omicron (January 2022–May 2023)**				
**Total**	**308**	**37.2**	**231**	**93.2**
Nonvaccinated HCWs	30	9.7	29	12.5
HCWs vaccinated with the primary series	139	45.1	130	56.3
Median number of days since immunization (IQR)	352 (241–384)	-	365 (238–519)	-
HCWs vaccinated with the first booster dose	139	45.1	72	31.2
Median number of days since revaccination (IQR)	127 (98–257)	-	169 (99–361)	-

SD, standard deviation; IQR, interquartile range; * percent of reinfections.

**Table 3 vaccines-12-00682-t003:** The incidence density of SARS-CoV-2 reinfection, in those vaccinated with the primary schedule and vaccinated with the first booster HCWs by demographic characteristics.

Characteristics	Reinfections during Omicron in HCWs Vaccinated with the Primary Schedule			Reinfections during Omicron in HCWs Vaccinated with the First Booster			Incidence Rate Ratio (95% CI)	*p*-Value
	N	P-d	Rate × 10,000 (95% CI)	N	P-d	Rate × 10,000 (95% CI)		
Total	130	179,208	7.3 (6.1; 8.6)	72	130,371	5.5 (4.3; 6.9)	0.8 (0.6; 1.0)	0.062
**Gender**								
Female	105	154,864	6.8 (5.5; 8.2)	63	106,242	5.9 (4.6; 7.6)	1.1 (0.8; 1.6)	0.402
Male	25	24,344	10.3 (6.7; 15.2)	9	24,129	3.7 (1.7; 7.1)	2.8 (1.2; 6.7)	0.006
**Age**								
≤29	6	16,179	3.7 (1.4; 8.1)	8	15,398	5.2 (2.2; 10.2)	0.7 (0.2; 2.3)	0.545
30–39	36	41,502	8.7 (6.1; 12.0)	17	31,329	5.4 (3.2; 8.7)	1.6 (0.9; 3.0)	0.108
40–49	49	68,750	7.1 (5.3; 9.4)	23	41,592	5.5 (5.5; 8.3)	1.3 (0.7; 2.4)	0.405
≥50	39	52,777	7.4 (5.3; 10.1)	24	42,052	5.7 (3.7; 8.5)	1.3 (0.8; 2.2)	0.322
**Job category**								
Nurses	60	79,950	7.5 (5.7; 9.7)	24	45,965	5.2 (3.3; 7.8)	1.4 (0.9; 2.4)	0.129
Physicians	15	16,179	9.3 (5.2; 15.3)	28	45,984	6.1 (4.0; 8.8)	1.5 (0.8; 2.9)	0.197
Healthcare auxiliary activities	36	51,822	6.9 (4.9; 9.6)	14	24,181	5.8 (3.2; 9.7)	1.2 (0.6; 2.4)	0.576
Other categories	19	31,257	6.1 (3.7; 9.5)	6	14,241	4.2 (1.5; 9.2)	1.4 (0.6; 4.4)	0.448
**Department**								
High Risk ^1^	102	119,918	8.5 (6.9; 10.3)	63	103,558	6.1 (4.7; 7.8)	1.4 (1.0; 1.9)	0.035
Low risk ^2^	28	59,290	4.7 (3.1; 6.8)	9	26,813	3.4 (1.5; 6.4)	1.4 (0.6; 3.4)	0.381

^1^ ICU, emergency, COVID-19 wards; ^2^ nonCOVID-19 wards, laboratory, other departments.

**Table 4 vaccines-12-00682-t004:** The incidence density of SARS-CoV-2 reinfection in HCWs vaccinated with the primary schedule by history of infection before and after the Omicron era.

Characteristics	History of Infection before January 2022			History of Infection after January 2022			Incidence Rate Ratio (95% CI)	*p*-Value
	N	P-d	Rate × 10,000 (95% CI)	N	P-d	Rate × 10,000 (95% CI)		
**HCWs Vaccinated with the ** **Primary Schedule**	115	108,570	10.6 (8.7; 12.7)	15	70,638	2.1 (1.2; 3.5)	5.0 (2.9; 9.2)	<0.001
**Gender**								
Female	95	94,359	10.1 (8.2; 12.3)	10	60,505	1.7 (0.8; 3.0)	6.1 (3.2; 13.1)	<0.001
Male	20	14,211	14.1 (8.6; 21.7)	5	10,133	4.9 (1.6; 11.5)	2.9 (1.0; 9.7)	0.026
**Age**								
≤29	5	7180	7.0 (2.3; 16.3)	1	8999	1.1 (0.0; 6.2)	6.3 (0.7; 296.4)	0.073
30–39	32	23,366	13.7 (9.4; 19.3)	4	18,136	2.2 (0.6; 5.6)	6.2 (2.2; 24.2)	<0.001
40–49	43	43,875	9.8 (7.1; 13.2)	6	24,875	2.4 (0.9; 5.3)	6.1 (2.2; 23.4)	<0.001
≥50	35	34,149	10.3 (7.1; 14.3)	4	18,628	2.1 (0.6; 5.5)	7.2 (2.7; 27.5)	<0.001
**Job category**								
Nurses	53	49,520	10.7 (8.0; 14.0)	7	30,430	2.3 (0.9; 4.7)	6.3 (0.7; 296.4)	0.073
Physicians	12	8863	13.5 (7.0; 23.7)	3	7316	4.1 (0.8; 12.0)	3.3 (0.9; 18.2)	0.051
Healthcare auxiliary activities	33	33,053	10.0 (6.9; 14.0)	3	18,769	1.6 (0.3; 4.7)	6.2 (2.0; 31.8)	<0.001
Other categories	17	17,134	9.9 (5.8; 15.9)	2	14,123	1.4 (0.2; 5.1)	7.0(1.7; 62.5)	0.002
**Department**								
High Risk ^1^	91	71,816	12.7 (10.2; 15.6)	11	48,102	2.3 (1.1; 4.1)	5.5 (3.0; 11.5)	<0.0001
Low risk ^2^	24	36,754	6.5 (4.2; 9.7)	4	22,536	1.8 (0.5; 4.5)	3.7 (1.3; 14.6)	0.0072

^1^ ICU, emergency, COVID-19 wards; ^2^ nonCOVID-19 wards, laboratory, other departments.

**Table 5 vaccines-12-00682-t005:** The incidence density of SARS-CoV-2 reinfection in HCWs vaccinated with the first booster by history of infection before and after the Omicron era.

Characteristics	History of Infection before January 2022			History of Infection after January 2022			Incidence Rate Ratio (95% CI)	*p*-Value
	N	P-d	Rate × 10,000 (95% CI)	N	P-d	Rate × 10,000 (95% CI)		
**HCWs Vaccinated with the First Booster**	**53**	**60,438**	8.8 (6.6; 11.5)	**19**	**69,933**	2.7 (1.6; 4.2)	3.2 (1.9; 5.8)	<0.001
**Gender**								
Female	45	51,578	8.7 (6.4; 11.7)	18	54,664	3.3 (2.0; 5.2)	2.6 (1.5; 4.9)	<0.001
Male	8	8860	9.0 (3.9; 17.8)	1	15,269	0.7 (0.0; 3.6)	13.8 (1.8; 611.8)	0.002
**Age**								
≤29	5	6155	8.1 (2.6; 19.0)	3	9243	3.2 (0.7; 9.5)	2.5 (0.5; 16.1)	0.223
30–39	13	11,646	11.2 (5.9; 19.1)	4	19,683	2.0 (0.6; 5.2)	5.5 (1.7; 23.1)	0.002
40–49	15	16,973	8.8 (4.9; 14.6)	8	24,619	3.3 (1.4; 6.4)	2.7 (1.1; 7.4)	0.021
≥50	20	25,664	7.8 (4.8; 12.0)	4	16,388	2.4 (0.7; 6.2)	3.2 (1.1; 12.8)	0.022
**Job category**								
Nurses	19	23,464	8.1 (4.9; 12.6)	5	22,501	2.2 (0.7; 5.2)	3.6 (1.3; 12.5)	0.006
Physicians	17	18,213	9.3 (5.4; 14.9)	11	27,771	4.0 (2.0; 7.1)	2.4 (1.0; 5.6)	0.027
Healthcare auxiliary activities	11	12,760	8.6 (4.3; 15.4)	3	11,421	2.6 (0.5; 7.7)	3.3 (0.9; 18.3)	0.056
Other categories	6	6001	10.0 (3.7; 21.8)	0	8240	0 (0;0)	0 (0;0)	-
**Department**								
High Risk ^1^	46	45,773	10.1 (7.4; 13.4)	17	57,785	2.9 (1.7; 4.7)	3.4 (1.9; 6.4)	<0.001
Low risk ^2^	7	14,665	4.8 (1.9; 9.8)	2	12,148	1.6 (0.2; 5.9)	2.9 (0.6; 28.6)	0.182

^1^ ICU, emergency, COVID-19 wards; ^2^ nonCOVID-19 wards, laboratory, other departments.

## Data Availability

The datasets generated and analyzed during the current study are available from the corresponding author upon reasonable request.

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
