# Peer review of "Hybrid Immunity and the Incidence of SARS-CoV-2 Reinfections during the Omicron Era in Frontline Healthcare Workers"

_vaccines, 2024, doi:10.3390/vaccines12060682_

Round 1

Reviewer 1 Report

Comments and Suggestions for Authors

In the manuscript entitled ‘Hybrid immunity and the incidence of SARS-CoV-2 reinfection during the Omicron era, in Healthcare Workers from a frontline hospital in Romania’ the authors report cases of infection/reinfection related to the vaccination history of 829 healthcare workers (HCWs). These data are important to understand the rule of the hybrid immunization against SARS-CoV-2 infection.

In literature, many articles describe this topic, and in my opinion, the authors should stress their purposes already in the introduction section.

In detail, the authors should better explain:

-          Why did they decide to disregard the unvaccinated HCWs? This could give more value to the role and the importance of hybrid immunization.

-          As reported by the authors, this manuscript continues and complements their previous research. So, after the analysis of these data, in the discussion section, could the authors give a general overview of their studies and summarize the data in their complexity? In the introduction section, could the authors describe how this study is different from the others and explain its importance in this field compared to t the other already published?

There are typos and SARS-CoV-2 was misspelled throughout the text.

Author Response

Dear Reviewer,

Thank you very much for the helpful suggestions. We consider that they consistently improved our script.

Sincerely, Maria Dorina Crăciun

Reviewer 2 Report

Comments and Suggestions for Authors

The study analyzed reinfection rates among subgroups of health care workers according to infection history, vaccination status and dominant virus variant, revealing a higher reinfection density among those with primary vaccination compared to those with a booster.

 I make the following suggestions to the authors:

 The title seeks to cover too much ground at once (hybrid immunity, incidence of reinfection, specific population and area). I advise concentrating on the primary research issue or finding.

The capitalization is inconsistent (for example, "Healthcare Workers" should actually be written "healthcare workers").

When first used in a text, acronyms should be explained (e.g. COVID-19, SARS-CoV-2). Also, the use of the HCW is excessive in abstract.

For better readability of some sentences in the abstract, simplify their structure.

Some abbreviations (e.g. RBD, VOI, VUM) does not make sense because they are used only once in the text. At the second mention in the text, only the acronym is used (e.g. NIPH).

It is recommended to use "B.1.1.1.7 lineage" without spaces.

To help readers understand the duration of the data collection period, it is important to provide precise information about the study's duration ("from January 2022 to May 31, 2023").

The format of percentages should be constant (e.g., "308; 37.2%" vs. "248 (29.9%)").

A typographical error (such as “HCWS” instead of “HCWs”).

The discussion states that reinfections were more common in high-risk departments and that vaccination with the first and primary vaccine does not reduce the incidence of reinfections. This is contradictory and needs to be clarified. Regardless of vaccination strategy, are high-risk departments naturally more prone to reinfection?

The statement that the study revealed a higher rate of reinfections compared to other studies is not sufficiently established. There is no comparison analysis or explanation of methodology and contexts of other studies, which is important to substantiate this remark.

Although the discussion implies an increase in COVID-19 cases during the Omicron variant, a comprehensive analysis is absent. How did Omicron particularly alters transmission dynamics among HCWs? What are the consequences of this increased transmissibility? Does that suggest that HCWs did not comply with the protective measures established by the authorities?

Assess the data and consider other reasons for the group differences and trends.

The comparison between health workers who were vaccinated with the primary scheme and those who received the first booster should provide more detailed statistical evidence in the text.

Hybrid immunity is discussed briefly in the text, despite mentioning its importance. Further discussion of its implications for public health workers would be beneficial.

Provide detailed comparisons with specific data when discussing the effectiveness of different vaccination programs or boosters.

The text does not sufficiently link previous research showing mechanisms related to post-vaccination antibody levels to the present study. These studies seem irrelevant and unrelated to the study unless the authors show better this link.

For a more comprehensive and impactful conclusion I suggest discussing other relevant factors influencing reinfection rates or broader implications for public health strategies.

Comments on the Quality of English Language

The English language is correctly used, but some sentences should be simplified to be more easily understood by readers.

Author Response

(The authors gave the same response as above.)
